# GeoMask3D: Geometrically Informed Mask Selection for Self-Supervised Point Cloud Learning in 3D

**Ali Bahri**[1,2][*]**, Moslem Yazdanpanah**[1,2]**, Mehrdad Noori**[1,2]**, Milad Cheraghalikhani**[1,2]**, Gustavo Adolfo Vargas Hakim**[1,2]**, David Osowiechi**[1,2]**, Farzad Beizaee**[1,2] **Ismail Ben Ayed**[1,2]**, Christian Desrosiers**[1,2]

[1]**École de technologie supérieure (ÉTS)**
[2]**International Laboratory on Learning Systems (ILLS)**

**Reviewed on OpenReview:** `https://openreview.net/forum?id=Yk7GUlJwGa`

## Abstract

We introduce a novel approach to self-supervised learning for point clouds, employing a geometrically informed mask selection strategy called GeoMask3D (GM3D) to boost the efficiency of Masked Auto Encoders (MAE). Unlike the conventional method of random masking, our technique utilizes a teacher-student model to focus on intricate areas within the data, guiding the model's focus toward regions with higher geometric complexity. This strategy is grounded in the hypothesis that concentrating on harder patches yields a more robust feature representation, as evidenced by the improved performance on downstream tasks. Our method also presents a feature-level knowledge distillation technique designed to guide the prediction of geometric complexity, which utilizes a comprehensive context from feature-level information. Extensive experiments confirm our method's superiority over State-Of-The-Art (SOTA) baselines, demonstrating marked improvements in classification, segmentation, and few-shot tasks. Code is available on *our GitHub*.

## 1 Introduction

The advent of large-scale 3D datasets (Deitke et al., 2023; Slim et al., 2023) has propelled research on deep learning for point clouds, leading to notable improvements in complex 3D tasks. However, the time-consuming nature of data collection, compounded by the complexity of 3D view variations and the mismatch between human perception and point cloud representation, significantly hinders the development of effective deep networks for this type of data (Xiao et al., 2023). In response to this challenge, Self-Supervised Learning (SSL) has emerged as a promising solution, facilitating the learning of representations without relying on manual annotations. SSL not only circumvents the issues of costly and error-prone labeling but also improves the model's generalization ability, offering a pivotal advancement in the field of point cloud-based deep learning (Fei et al., 2023).

MAEs, as simple yet effective self-supervised learners, have gained prominence by learning to recover masked parts of data. This approach has significantly advanced NLP models and has resulted in exceptional vision-based representation learners (Kenton & Toutanova, 2019; He et al., 2022) when applied to vision

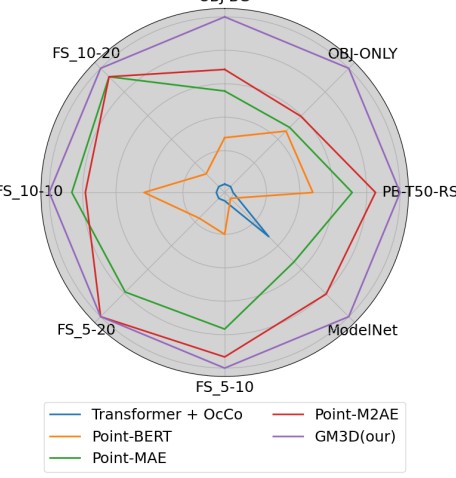

Figure 1: A relative comparison of the SOTA point cloud MAE methods on different tasks. Here, the center and the outer circles represent the lowest and highest values on each task, respectively.

---

[*]Correspondence to ali.bahri.1@ens.etsmtl.ca

tasks. Building on their success, MAEs have recently been adapted for point cloud representation learning, leading to SOTA methods including MaskPoint (Liu et al., 2022), Point-Bert (Yu et al., 2022), Point-MAE (Pang et al., 2022), Point-M2AE (Zhang et al., 2022), I2P-MAE (Zhang et al., 2023), and MAE3D (Jiang et al., 2023). However, these methods share a common limitation stemming from the random masking strategy they use, where masked regions of the point cloud are selected arbitrarily without taking into account their informativeness. As demonstrated in recent work on Masked Image Modeling (MIM) (Wang et al., 2023), employing a selection strategy that prioritizes informative regions over background areas can significantly enhance the robustness of the learned representation. While such a strategy has shown promising results for image processing, its application to point clouds has not been explored so far.

To bridge this gap, we study the use of a targeted masking strategy for point clouds within the MAE framework, applicable across both single and multi-scale methods. We introduce GeoMask3D (GM3D), a novel geometrically-informed mask selection strategy for object point clouds. Due to the lack of background data in the object's point clouds, GM3D enables models to concentrate on more complex areas, such as canonical ones with higher connections to the other areas, and pay less attention to the geometrically simple areas like smooth surfaces, as depicted in Fig. 2. To showcase the effectiveness of our method, we integrate it into the pretraining process of both single-scale Point-MAE and multi-scale Point-M2AE, the leading MAE methods for point clouds. As illustrated in Fig. 1, our method exhibits notable enhancements over the earlier SOTA approaches across a range of challenging tasks.

To the best of our knowledge, this represents the first attempt to implement a masking strategy for point clouds, independent of additional modalities such as multi-view images. Our contributions are summarized as follows:

1. We propose a novel masking approach for point cloud MAEs, which selects patches based on their geometric complexity rather than selecting them randomly. This approach employs an easy-to-hard curriculum learning strategy where the ratio of patches selected using geometric complexity is gradually increased during training.

2. We also introduce a feature-level knowledge distillation technique to further guide the prediction of geometric complexity. Instead of relying on the noisy and incomplete information of 3D points, this efficient technique transfers latent features from a frozen teacher model, encoding higher-level information on the geometry, to the student model learning the point cloud representation.

3. We integrate these mechanisms into the pretraining process of single-scale Point-MAE and multi-scale PointM2AE, both of which are SOTA point cloud MAEs, significantly enhancing their performance in diverse downstream tasks.

## 2 Related Works

**Point Cloud Learning.** PointNet (Qi et al., 2017a) established point cloud processing as a key method in 3D geometric data analysis by addressing the permutation issue of point clouds with a max-pooling layer. To further enhance performance and capture both local and global features, PointNet++ (Qi et al., 2017b) introduced a hierarchical structure, expanding the receptive fields of its kernels recursively for improved results over PointNet. Another study (Jaritz et al., 2019) focused on point cloud scene processing, where multi-view image features are combined with point clouds. In this study, 2D image features are aggregated into 3D point clouds and a point-based network fuses these features in 3D space for semantic labeling, demonstrating the substantial benefits of multi-view information in point cloud analysis.

**MAE for Representation Learning.** Leveraging the success of MAEs in text and image modalities, Point-BERT (Yu et al., 2022) introduced an approach inspired by BERT (Devlin et al., 2018) adapting Transformers to 3D point clouds. This approach creates a Masked Point Modeling task, partitioning point clouds into patches and using a Tokenizer with a discrete Variational AutoEncoder (dVAE) to produce localized point tokens, with the goal of recovering original tokens at masked points. Similarly, Point-GPT (Chen et al., 2024) introduced an auto-regressive generative pretraining (GPT) approach to address the unordered nature and low information density of point clouds. ACT (Dong et al., 2022) proposed a cross-modal knowledge

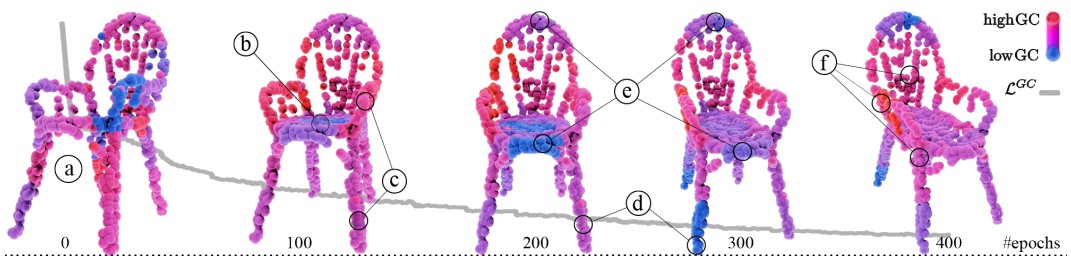

Figure 2: Visualization of estimated Geometric Complexity (GC) progression throughout training is depicted. The color spectrum denotes GC, ranging from low (Blue) to high (Red). GC values are normalized per object to reflect relative complexity across patches within each object's point cloud. As training progresses (from left to right), initial GC rankings display a random distribution (a). After 100 epochs, the model learns to assign lower complexity rankings to smooth areas (b) and higher rankings to complex regions (c). Through GC guided masking, the model increasingly focuses on complex areas from epochs 200 to 300, resulting in a reduction of GC ranking (d) and smoothing of the complexity ranking distribution, accompanied by a decrease in total complexity loss $\mathcal{L}^{GC}$ (e). Eventually, the model converges to a low $\mathcal{L}^{GC}$ value, consistently targeting canonical patches while maintaining a smoother GC distribution (f).

transfer method using pretrained 2D or natural language Transformers as teachers for 3D representation learning. MaskPoint (Liu et al., 2022), a discriminative mask pretraining framework for point clouds, represents the point cloud with discrete occupancy values and performs binary classification between object points and noise points, showing resilience to point sampling variance. Point-MAE (Pang et al., 2022) adapted MAE-style pretraining to 3D point clouds, employing a specialized Transformer-based autoencoder to reconstruct masked irregular patches and demonstrating strong generalization in various tasks. Following this, MAE3D (Jiang et al., 2023) used a Patch Embedding Module for feature extraction from unmasked patches. Point-M2AE (Zhang et al., 2022) introduced a Multi-scale MAE framework with a pyramid architecture for self-supervised learning, focusing on fine-grained and high-level semantics. I2P-MAE (Zhang et al., 2023) further improved the self-supervised point cloud learning process by leveraging pretrained 2D models through an Image-to-Point transformation.

Our proposed method, which can be integrated into any point cloud MAE architecture, differs from previous approaches like Point-MaskPoint, MAE, MAE3D and Point-M2AE that are based on random patch selection. Moreover, unlike recent point cloud learning approaches such as I2P-MAE (Zhang et al., 2023), which rely on image information and 2D backbones, our method only requires 3D coordinates as input.

## 3 Method

### 3.1 Preliminaries

**Masked Auto Encoders.** Autoencoders use an encoder $\mathcal{E}$ to map an input $X$ to a latent representation $Z = \mathcal{E}(X)$ and a decoder to reconstruct the input as $\hat{X} = \mathcal{D}(Z)$. Masked Auto Encoders (MAE) are a special type of autoencoder that receive a masked-patch input composed of a set of visible patches (with positional encoding) $X^v$ and the index set of masked patches $M$, and reconstruct the masked patches as follows:

$$[X^v, \hat{X}^m] = \text{MAE}(X^v, M) = \mathcal{D}\big([\mathcal{E}(X^v), T_M]\big) \tag{1}$$

The encoder $\mathcal{E}$ and decoder $\mathcal{D}$ are both transformer-based networks. The encoder only transforms the visible patches to their latent representations $Z^v = \mathcal{E}(X^v)$. On the other hand, the decoder takes as input $Z^v$ and a set of tokens $T_M = \big\{ \big(t_{mask}, E_{pos}(i)\big) \,|\, i \in M \big\}$ where $t_{mask}$ is a global learnable mask token and $E_{pos}(i)$ is the positional embedding of masked patch $i \in M$. Let $N = N^v + N^m$ be the number of patches in the input, with $N^v = |X^v|$ and $N^m = |M|$, the masking ratio is defined as $m^{ratio} = N^m/N$.

**MAE for Point Clouds.** A 3D point cloud $P$ is a set of $N^p$ points $p_j \in \mathbb{R}^3$. For this type of data, patches correspond to possibly overlapping subsets of $K$ points in $P$. While there are various ways to generate

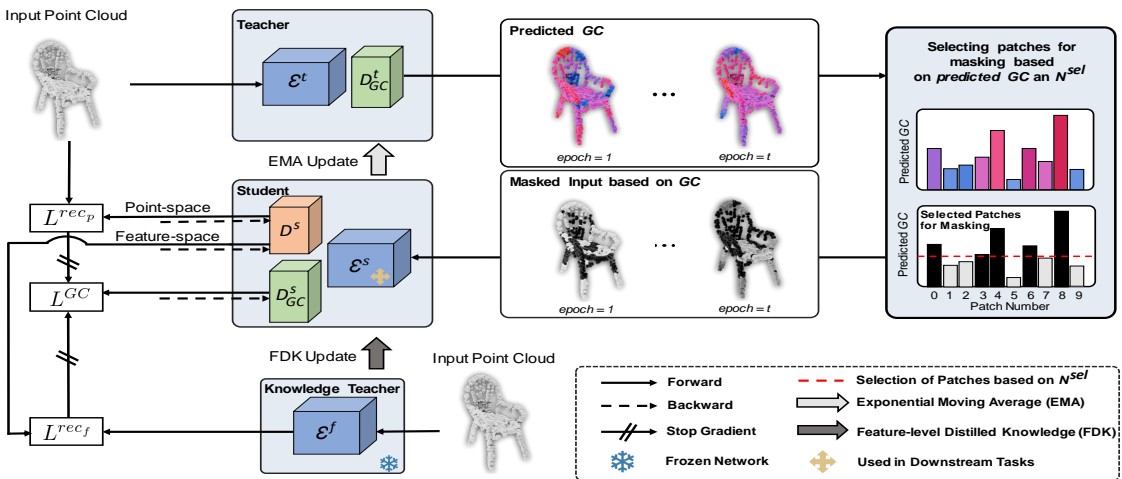

Figure 3: Overview of the GeoMask3D (GM3D) method for self-supervised representation learning in point clouds. The Teacher network predicts Geometric Complexity (GC), and patches with the highest GC, denoted by $N^{sel}$, are selected for masking. The Student network is then trained to reconstruct these masked tokens while simultaneously learning GC through the loss $\mathcal{L}^{GC}$. The reconstruction loss is defined as $\mathcal{L}^{rec} = \mathcal{L}^{rec_p} + \mathcal{L}^{rec_f}$. The Teacher network's weights are updated using the Exponential Moving Average (EMA) of the Student's weights, while the Knowledge Teacher remains frozen and is used for generating encoder features essential for the Student's training with $\mathcal{L}^{rec_f}$.

patches from a point cloud, we follow the strategy employed by several related methods (Pang et al., 2022; Zhang et al., 2022) where each patch $x_i$ is defined as a center point $c_i$ and the set $P_i \subset P$ of K-Nearest Neighbours (KNN) to this center. To uniformly represent the whole point cloud, patch centers $c_i$ are obtained using a Farthest Point Sampling (FPS) algorithm, where a first center is randomly chosen from $P$ and then the next one is selected as the point in $P$ furthest away from previously selected centers. Assuming that $c_i$ is included as the first element of its nearest-neighbor list $P_i$, we can represent the patchified version of the point cloud as a tensor $X \in \mathbb{R}^{N \times K \times 3}$.

In autoencoders for images, a pixel-wise L2 reconstruction loss is typically used for training the MAE. In our case, since patches $x_i \in X$ are points sets, we instead employ the Chamfer distance to measure the reconstruction error $\mathcal{L}^{rec_p}$ of masked patches:

$$\mathcal{L}^{rec_p} = \frac{1}{N^m} \sum_{i=1}^{N^m} \text{Chamfer}(x_i^m, \hat{x}_i^m), \qquad (2)$$

where the Chamfer distance between two sets of points $S$ and $S'$ is defined as

$$\text{Chamfer}(S, S') = \sum_{p \in S} \min_{p' \in S'} \|p - p'\|_2^2 + \sum_{p' \in S'} \min_{p \in S} \|p - p'\|_2^2. \qquad (3)$$

In the next section, we build on these definitions and present our GeoMask3D method for self-supervised point cloud representation learning.

## 3.2 GeoMask3D

In the originally proposed MAE, masked patches are selected randomly during each iteration without considering the varying impacts that different patches may have on the training process. This random masking approach may not be efficient, as patches in a point cloud can exhibit varying levels of Geometric Complexity (GC), which pose different degrees of challenge to the learning network. Inspired by the principles of human learning—where repeatedly tackling challenging tasks enhances performance over time—we propose prioritizing geometrically complex patches during the pre-training phase of a MAE network. A MAE network

can achieve more efficient and effective learning by shifting from random masking to a strategy focusing on complex patches.

This strategy raises a fundamental question: what is Geometric Complexity (GC), and how can it be measured? We define GC for a patch as the relative difficulty of reconstructing that patch using an MAE network. Specifically, a patch is considered complex if the MAE network demonstrates difficulty in reconstructing it, as indicated by higher reconstruction loss $\mathcal{L}^{rec_p}$.

To this end, we propose GeoMask3D (GM3D) as a modular component that integrates with any point cloud MAE backbone. This component is incorporated into the pretraining phase of a chosen method, shifting from a basic naive random masking approach to a selective focus on geometrically complex patches for masking. The architecture of GM3D employs an auxiliary head $\mathcal{D}_{GC}$ for predicting the geometric complexity $GC \in \mathbb{R}^N$ across the input patches, which is trained with a loss $\mathcal{L}^{GC}$.

A teacher-student framework is utilized to integrate GM3D with a target network. We denote as $\text{GM3D}^s = (\mathcal{E}^s, \mathcal{D}^s, \mathcal{D}^s_{GC})$ the student and as $\text{GM3D}^t = (\mathcal{E}^t, \mathcal{D}^t, \mathcal{D}^t_{GC})$ the teacher, both of them having their own encoder, decoder and auxiliary head. In line with (He et al., 2020), we apply a momentum update method to maintain a consistent teacher, updating it in each iteration,

$$\text{GM3D}^t = \mu \cdot \text{GM3D}^t + (1-\mu) \cdot \text{GM3D}^s \tag{4}$$

where $\mu$ represents the momentum coefficient. Both networks predict the GC based on the patch's informational content, as elaborated in Section 3.2.1. We employ the prediction of GC in the masking strategy of the method during its pretraining stage. The GC of the student network ($\text{GC}^s$) is predicted based on the masked input $X^v$, while the GC of the teacher ($\text{GC}^t$) is calculated in inference mode using the complete input $X$:

$$GC^a = \begin{cases} \mathcal{D}^a_{GC}(\mathcal{E}(X)), & \text{if } a = t \ \ (teacher) \\ \mathcal{D}^a_{GC}\big([\mathcal{E}(X^v), T_M]\big), & \text{if } a = s \ \ (student) \end{cases} \tag{5}$$

The overview of our method for self-supervised representation learning in the point cloud is depicted in Fig. 3. The GeoMask3D (GM3D) approach involves three interconnected steps, which will be explained in the following sections. Additionally, we provide a detailed explanation of the Knowledge-Distillation-Guided GC strategy in Section 3.3.

### 3.2.1 Prediction of Geometric Complexity (GC)

During this stage, our goal is to evaluate GC of each patch in $X^m$, relative to the others within the same set. We achieve this by using a Dense Relation Comparison (DRC) loss (Wang et al., 2023) which enforces the GC of masked patch pairs $(k, l)$, predicted by the student (i.e., $GC^s_k$ and $GC^s_l$), to follow the same relative order as their loss values $\mathcal{L}^{rec}_k$, $\mathcal{L}^{rec}_l$:

$$\mathcal{L}^{GC} = \sum_{k=1}^{N^m} \sum_{\substack{l=1 \\ l \neq k}}^{N^m} \mathcal{I}^+_{kl} \log\big(\sigma(GC^s_k - GC^s_l)\big) - \mathcal{I}^-_{kl} \log\big(1 - \sigma(GC^s_k - GC^s_l)\big) \tag{6}$$

where $\mathcal{I}^+_{kl} = \mathbb{1}(\mathcal{L}^{rec}_k > \mathcal{L}^{rec}_l)$, $\mathcal{I}^-_{kl} = \mathbb{1}(\mathcal{L}^{rec}_k < \mathcal{L}^{rec}_l)$, $\sigma(\cdot)$ is the sigmoid function, and $\mathcal{L}^{rec}$ is detailed in Section 3.3.

This loss function enforces consistency between the predicted $GC^s$ values and $\mathcal{L}^{rec}$ as the *ground truth*, effectively guiding the student model to learn a meaningful ranking of geometric complexities for the masked patches. By comparing all pairs of patches, the loss ensures a robust evaluation of relative complexity within $X^m$.

### 3.2.2 Geometric-Guided Masking

Patches with a high GC score are typically those that the model struggles to reconstruct accurately (see Fig. 2). This difficulty often stems from their complex geometry, compounded by the absence of color and

background information. While choosing those patches for masking might seem straightforward, there are two challenges to this approach. First, during training, the GC is evaluated by the student for masked patches $X^m$, yet we need to pick candidate patches from the entire set. Second, the student's GC estimation can be noisy, making the training unstable. We address both these challenges by instead selecting patches based on the teacher's score (GC$^t$). Thus, at each iteration, the teacher predicts the GC for all patches in $X$, including unmasked ones. Thanks to the momentum update of Fig. 4, the teacher's predictions are more consistent across different training iterations.

### 3.2.3   Curriculum Mask Selection

During the initial phases of training, the model may struggle to reconstruct fine details and is often overwhelmed by the complexity of the point cloud structure. To mitigate this problem, we follow a curriculum easy-to-hard mask selection strategy by starting from pure random masking at the initial training epoch and gradually increasing the portion of geometric-guided masking until the maximum epoch $e_{max}$. Let $A \in [0,1]$ be the maximum ratio of patches that can be selected using GC$^t$. At each epoch $e_t$, we select the $N^{sel} = \lfloor e_t / e_{max} \times A \times N^m \rfloor$ patches with highest GC$^t$ value, and the remaining $N^m - N^{sel}$ ones are selected randomly based on a uniform distribution.

### 3.3   Knowledge-Distillation-Guided GC

Instead of relying exclusively on point geometry, our approach employs a knowledge distillation strategy to also learn from latent features. This strategy involves transferring geometric knowledge from a frozen teacher network $F = (\mathcal{E}^f, \mathcal{D}^f)$ that processes the full set of patches to the student GM3D$^s$ observing unmasked patches. This encourages the student GM3D$^s$ to replicate the feature activations of the knowledge teacher $F$, indirectly learning from the full structure of data. This unique setup enables the student network to benefit from the global geometric context provided by the teacher network, which is constructed from the complete point cloud. As a result, this process facilitates the learning of robust and meaningful representations, which improve performance on downstream tasks. The complexity of patches in the feature space is determined by employing the Mean Square Error loss between the output of $\mathcal{E}^f$ and the output of $\mathcal{D}^s$ before converting back to point space:

$$\mathcal{L}^{rec_f} = \frac{1}{N^m} \sum_{i=1}^{N^m} \left\| \mathcal{E}^f(X)_i - \mathcal{D}^s([\mathcal{E}^s(X^v), T_M])_i \right\|^2 \tag{7}$$

This loss, combined with the Chamfer loss $\mathcal{L}^{rec_p}$ applied in the point space, serves as the ground-truth loss $\mathcal{L}^{rec}$ for the prediction of GC:

$$\mathcal{L}^{rec} = \mathcal{L}^{rec_p} + \mathcal{L}^{rec_f} \tag{8}$$

The total training loss $\mathcal{L}$ is calculated as

$$\mathcal{L} = \alpha \mathcal{L}^{GC} + \beta \mathcal{L}^{rec_p} + \gamma \mathcal{L}^{rec_f} \tag{9}$$

where $\alpha$, $\beta$, and $\gamma$ are hyper-parameters.

## 4   Experiments

Several experiments are carried out to evaluate the proposed method. First, we pretrain both Point-MAE and Point-M2AE networks utilizing our GM3D approach on the ShapeNet (Chang et al., 2015) training dataset. Moreover, we assess the performance of these pretrained models across a range of standard benchmarks, such as object classification, few-shot learning, and part segmentation. It is important to note that, to maintain a completely fair comparison, we exclusively utilize the encoder of the student network for downstream tasks, ensuring it is identical to the encoder used in the method of interest.

In our approach, we adopt network configurations consistent with those used in the Point-MAE and Point-M2AE models to guarantee a fully fair comparison, notably using masking ratios of 60% for Point-MAE and 80% for Point-M2AE. This involves the technique of dividing point clouds into patches, along with employing the KNN algorithm with predetermined parameters for consistent patch uniformity. While our

autoencoder architecture, including the configuration of Transformer blocks in both encoder and decoder, generally follows the patterns established in these models, we have uniquely tailored the decoder's design specifically for the GC estimation purposes. Moreover, the specifics of our network's hyperparameters for the pretraining and fine-tuning phases are comprehensively detailed in the Supplementary Materials.

## 4.1 Pretraining Setup

We adopt the ShapeNet dataset (Chang et al., 2015) for the pretraining of our technique, in line with the practices established by Point-MAE and Point-M2AE. This dataset, known for its diverse and extensive collection of 3D models across various categories, provides a robust basis for training and evaluation. It contains 57,448 synthetic 3D shapes of 55 categories.

After this pretraining phase, we assess the quality of 3D representations produced by our approach through a linear evaluation on the ModelNet40 dataset (Wu et al., 2015). We extract 1,024 points from every 3D model in ModelNet40 and then pass them through our encoder, which remains unchanged during this phase to preserve the learned features. The linear evaluation is performed by a Support Vector Machine (SVM) fitted on these features. This classification performance is quantified by the accuracy metrics detailed in Table 1. The results clearly indicate that our technique, when applied to Point-MAE and Point-M2AE, enhances the network's performance.

## 4.2 Downstream Tasks

**Object Classification on Real-World Dataset.** In self-supervised learning for point clouds, it is crucial to create a model that exhibits strong generalization abilities across various scenarios. The ShapeNet dataset, which is favored for pretraining, contains clean, isolated object models, lacking any intricate scenes or background details. Inspired by this limitation, and building on prior approaches, we put our methods to the test on the ScanObjectNN dataset (Uy et al., 2019), a more demanding dataset that represents about 15,000 real-world objects across 15 categories. This dataset presents a realistic challenge, with objects that are embedded in cluttered backgrounds, making it an ideal benchmark for assessing our model's robustness and generalization in real-world scenarios.

We carry out tests on three different variants: OBJ-BG, OBJ-ONLY, and PBT50-RS. It is important to note that we do not employ any voting techniques or data augmentation during the testing phase. The outcomes of these experiments can be found in Table 5. These results demonstrate that integrating the GM3D module with Point-MAE and Point-M2AE significantly boosts their object classification accuracy on this dataset. These findings underscore our method's effectiveness in complex real-world scenarios.

**Object Classification on Clean Objects Dataset.** For the task of object classification on the ModelNet40 dataset(Wu et al., 2015), we evaluated our pretrained models using the same protocols and configurations as the Point-MAE approach. ModelNet40, featuring 12,311 pristine 3D CAD models across 40 categories, was divided into a training set of 9,843 models and a testing set of 2,468 models, adhering to established norms. Throughout the training, we employed common data augmentation strategies, including random scaling and shifting. To ensure fair comparisons, the standard voting method (Liu et al., 2019) was also applied during the testing phase. According to Table 4, integrating our GM3D module with Point-MAE has yielded a classification accuracy of 94.20%, which surpasses the performance of the standalone Point-MAE and even the more complex Point-M2AE on this dataset.

**Few-shot Learning.** Following the protocols of earlier studies (Yu et al., 2022; Sharma & Kaul, 2020; Wang et al., 2021), we conduct few-shot learning experiments on ModelNet40(Wu et al., 2015), using an $n$-way, $m$-shot configuration. Here, $n$ is the number of classes randomly chosen from the dataset, and $m$ is the count of objects randomly selected for each class. The $n \times m$ objects are utilized for training. During the test phase, we randomly sample 20 additional unseen objects from each of the $n$ classes for evaluation.

The results of our few-shot learning experiments are summarized in Fig. 6. In this highly saturated benchmark, the combination of the GM3D module exhibits outstanding performance across all tested scenarios. It is worth noting that I2P-MAE(Zhang et al., 2023) which *additionally benefits from multiple 2D views* provides

Table 1: **Linear evaluation on Model-Net40 (Wu et al., 2015) by SVM.**

| Method | SVM |
|---|---|
| MAP-VAE (Wang et al., 2019) | 88.4 |
| VIP-GAN (Guo et al., 2021) | 90.2 |
| DGCNN + Jiasaw (Yu et al., 2022) | 90.6 |
| DGCNN + OcCo (Yu et al., 2022) | 90.7 |
| DGCNN + CrossPoint (Yu et al., 2022) | 91.2 |
| Transformer + OcCo (Yu et al., 2022) | 89.6 |
| Point-BERT (Yu et al., 2022) | 87.4 |
| Point-MAE (Pang et al., 2022) | 91.05 |
| **Point-MAE + GM3D** | **92.30** |
| Point-M2AE (Zhang et al., 2022) | 92.90 |
| **Point-M2AE + GM3D** | **93.15** |

Table 2: **Ablation study on different maximum hard patch ratios ($A$).** The highest performance is observed at 50%, where the OBJ-ONLY score reaches 90.36%.

| Model | $A$ | OBJ-ONLY |
|---|---|---|
| Original Point-MAE | 0 | 88.29 |
| Point-MAE+GM3D | 0.4 | 89.67 |
| Point-MAE+GM3D | **0.5** | **90.36** |
| Point-MAE+GM3D | 0.7 | 89.84 |

Table 3: **Part segmentation on ShapeNet-Part (Yi et al., 2016).** $mIoU_c$ (%) and $mIoU_i$ (%) denote the mean IoU across all part categories and all instances in the dataset, respectively. § represents self-supervised pertaining.

| Method | $mIoU_c$ | $mIoU_i$ |
|---|---|---|
| PointNet (Qi et al., 2017a) | 80.39 | 83.70 |
| PointNet++ (Qi et al., 2017a) | 81.85 | 85.10 |
| DGCNN (Wang et al., 2019) | 82.33 | 85.20 |
| Transformer (Yu et al., 2022) | 83.42 | 85.10 |
| §Transformer + OcCo (Yu et al., 2022) | 83.42 | 85.10 |
| §Point-BERT (Yu et al., 2022) | 84.11 | 85.60 |
| §I2P-MAE (Zhang et al., 2023) | 85.15 | 86.76 |
| §Point-GPT-S (Chen et al., 2024) | 84.10 | 86.2 |
| §ACT (Dong et al., 2022) | 84.66 | 86.16 |
| §Point-MAE (Pang et al., 2022) | 84.19 | **86.10** |
| §**Point-MAE + GM3D** | **84.49** | 86.04 |
| §Point-M2AE (Zhang et al., 2022) | 84.86 | 86.51 |
| §**Point-M2AE + GM3D** | **84.91** | **86.52** |

Table 4: **Linear evaluation on Model-Net40 (Wu et al., 2015).** 'points' and 'Acc' denote the number of points for training and overall accuracy. § represents self-supervised pretraining.

| Method | Points | Acc (%) |
|---|---|---|
| PointNet (Qi et al., 2017a) | 1k | 89.2 |
| PointNet++ (Qi et al., 2017a) | 1k | 90.5 |
| §SO-Net (Li et al., 2018a) | 5k | 92.5 |
| DGCNN (Wang et al., 2019) | 1k | 92.9 |
| Point Transformer (Zhao et al., 2021) | | 93.7 |
| Transformer (Yu et al., 2022) | 1k | 91.4 |
| §Transformer + OcCo (Yu et al., 2022) | 1k | 92.1 |
| §Point-BERT (Yu et al., 2022) | 1k | 93.2 |
| §Point-BERT (Yu et al., 2022) | 4k | 93.4 |
| §Point-BERT (Yu et al., 2022) | 8k | 93.8 |
| §Point-M2AE (Zhang et al., 2022) | 1k | 94.00 |
| §Point-GPT-S (Chen et al., 2024) | 1k | 94.00 |
| §ACT (Dong et al., 2022) | 1k | 93.5 |
| §I2P-MAE (Zhang et al., 2023) | 1k | 94.1 |
| §Point-MAE (Pang et al., 2022) | 1k | 93.80 |
| §**Point-MAE + GM3D** | **1k** | **94.20** |

only marginal improvements in results. Furthermore, Point-GPT (Chen et al., 2024) and ACT (Dong et al., 2022), despite being SOTA and complex methods, show only slight improvements compared to each other and other SOTA methods. Our findings highlight the effectiveness of our method as our Point-MAE+GM3D model has already achieved higher accuracy than single-scale Point-MAE and multi-scale Point-M2AE.

**Part Segmentation.** Our method's capacity for representation learning was assessed using the ShapeNet-Part dataset (Yi et al., 2016), which includes 16,881 objects across 16 different categories. In alignment with the approaches taken in prior studies (Qi et al., 2017a;b; Yu et al., 2022), we sampled 2,048 points from each object to serve as input.

For this highly competitive benchmark, our GM3D method achieves a slight improvement on both the Point-MAE and Point-M2AE networks, as detailed in Table 3. Considering that our approach exclusively utilizes 3D information, the observed improvement over methods like I2P-MAE(Zhang et al., 2023) that *supplement 3D with additional 2D data* is reasonable, especially considering the slight enhancements achieved by I2P-

Table 5: **Object classification on real-world ScanObjectNN dataset (Uy et al., 2019).** We evaluate our approach on three variants, among which PB-T50-RS is the hardest setting. Accuracy (%) for each variant is reported. ˢˢrepresents self-supervised pretraining.

| Method | OBJ-BG | OBJ-ONLY | PB-T50-RS |
|---|---|---|---|
| PointNet (Qi et al., 2017a) | 73.3 | 79.2 | 68.0 |
| SpiderCNN (Xu et al., 2018) | 77.1 | 79.5 | 73.7 |
| PointNet++ (Qi et al., 2017b) | 82.3 | 84.3 | 77.9 |
| DGCNN (Wang et al., 2019) | 82.8 | 86.2 | 78.1 |
| PointCNN (Li et al., 2018b) | 86.1 | 85.5 | 78.5 |
| BGA-DGCNN (Uy et al., 2019) | - | - | 79.7 |
| BGA-PN++ (Uy et al., 2019) | - | - | 80.2 |
| GBNet (Qiu et al., 2021) | - | - | 80.5 |
| PRANet (Cheng et al., 2021) | - | - | 81.0 |
| Transformer (Yu et al., 2022) | 79.86 | 80.55 | 77.24 |
| ˢˢTransformer-OcCo (Yu et al., 2022) | 84.85 | 85.54 | 78.79 |
| ˢˢPoint-BERT (Yu et al., 2022) | 87.43 | 88.12 | 83.07 |
| ˢˢI2P-MAE (Zhang et al., 2023) | 94.15 | 91.57 | 90.11 |
| ˢˢPoint-GPT-S (Chen et al., 2024) | 91.6 | 90.0 | 86.9 |
| ˢˢACT (Dong et al., 2022) | 93.29 | 91.91 | 88.21 |
| ˢˢPoint-MAE (Pang et al., 2022) | 90.02 | 88.29 | 85.18 |
| ˢˢ**Point-MAE + GM3D** | **93.11** | **90.36** | **88.30** |
| ˢˢPoint-M2AE (Zhang et al., 2022) | 91.22 | 88.81 | 86.43 |
| ˢˢ**Point-M2AE + GM3D** | **94.14** | **90.70** | **87.64** |

MAE. Furthermore, Point-GPT (Chen et al., 2024) and ACT (Dong et al., 2022), despite being SOTA and complex methods, show only slight improvements over each other and other SOTA methods. Based on the results of SOTA methods presented in Table 3, it is evident that this dataset is highly challenging and competitive. This highlights the effectiveness of our masking strategy in enhancing the understanding of detailed, point-wise 3D patterns.

Table 6: **Few-shot classification on ModelNet40**. We report the average accuracy (%) and standard deviation (%) of 10 independent experiments. ˢˢrepresents self-supervised pretraining.

| Method | 5-way | | 10-way | |
|---|---|---|---|---|
| | 10-shot | 20-shot | 10-shot | 20-shot |
| DGCNN (Wang et al., 2019) | $91.8_{\pm 3.7}$ | $93.4_{\pm 3.2}$ | $86.3_{\pm 6.2}$ | $90.9_{\pm 5.1}$ |
| ˢˢDGCNN + OcCo (Wang et al., 2021) | $91.9_{\pm 3.3}$ | $93.9_{\pm 3.1}$ | $86.4_{\pm 5.4}$ | $91.3_{\pm 4.6}$ |
| Transformer (Yu et al., 2022) | $87.8_{\pm 5.2}$ | $93.3_{\pm 4.3}$ | $84.6_{\pm 5.5}$ | $89.4_{\pm 6.3}$ |
| ˢˢTransformer + OcCo (Yu et al., 2022) | $94.0_{\pm 3.6}$ | $95.9_{\pm 2.3}$ | $89.4_{\pm 5.1}$ | $92.4_{\pm 4.6}$ |
| ˢˢPoint-BERT (Yu et al., 2022) | $94.6_{\pm 3.1}$ | $96.3_{\pm 2.7}$ | $91.0_{\pm 5.4}$ | $92.7_{\pm 5.1}$ |
| ˢˢI2P-MAE (Zhang et al., 2023) | $97.0_{\pm 1.8}$ | $98.3_{\pm 1.3}$ | $92.6_{\pm 5.0}$ | $95.5_{\pm 3.0}$ |
| ˢˢPoint-GPT-S (Chen et al., 2024) | $96.8_{\pm 2.0}$ | $98.6_{\pm 1.1}$ | $92.6_{\pm 4.6}$ | $95.2_{\pm 3.4}$ |
| ˢˢACT (Dong et al., 2022) | $96.8_{\pm 2.3}$ | $98.0_{\pm 1.4}$ | $93.3_{\pm 4.0}$ | $95.6_{\pm 2.8}$ |
| ˢˢPoint-M2AE (Zhang et al., 2022) | $96.8_{\pm 1.8}$ | $98.3_{\pm 1.4}$ | $92.3_{\pm 4.5}$ | $95.0_{\pm 3.0}$ |
| ˢˢPoint-MAE | $96.3_{\pm 2.5}$ | $97.8_{\pm 1.8}$ | $92.6_{\pm 4.1}$ | $95.0_{\pm 3.0}$ |
| ˢˢ**Point-MAE + GM3D** | $\mathbf{97.0_{\pm 2.5}}$ | $\mathbf{98.3_{\pm 1.3}}$ | $\mathbf{93.1_{\pm 4.0}}$ | $\mathbf{95.2_{\pm 3.6}}$ |

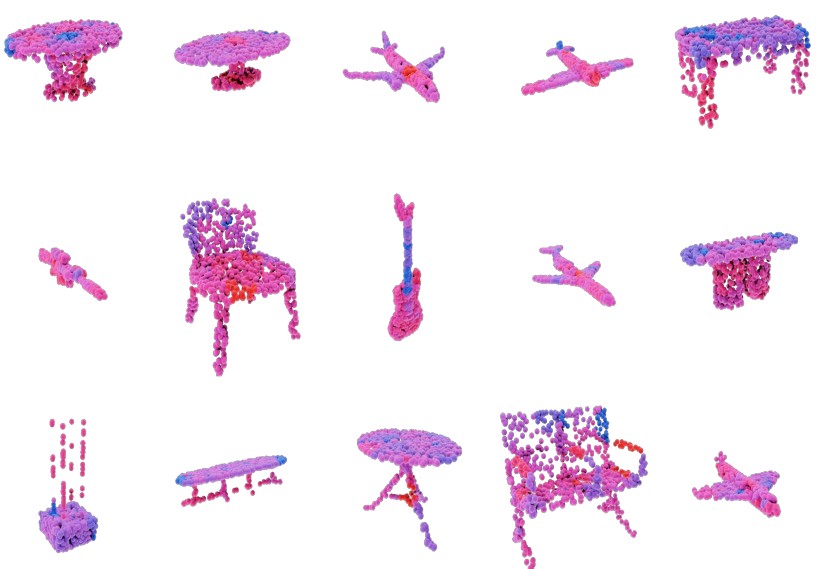

Figure 4: Visualization of GC values on diverse point clouds from the ShapeNet dataset (Chang et al., 2015).

### 4.3 Additional Visualization

**Geometric Complexity.** Fig. 4 illustrates the GC of randomly selected point clouds from the ShapeNet dataset. This illustration highlights the model's capability to assess GC at the patch level, where the red points denote areas of high GC and blue ones indicate areas of low GC. As mentioned in our methodology section, the model bases the masking process on the predicted GC of the patches. Consequently, patches representing regions with higher GC are preferentially masked. This strategic masking induces the model to focus intensively on intricate point cloud regions containing salient geometric information, thereby enhancing its overall performance in tasks requiring nuanced geometric understanding. It is important to note that the provided normalized GC scores are computed relative to the individual patches within each point cloud sample from the ShapeNet dataset. This normalization ensures that the GC scores are a reflection of the variation in complexity within a given sample, enabling the model to internally assess and compare different regions of the same point cloud.

### 4.4 Additional Analyses

**Pretraining Phase.** The convergence rates shown in Fig. 5 clearly highlight the efficiency of our proposed modules. Among the models, the Point-MAE+GM3D model stands out for its fast convergence, reaching high SVM accuracy with fewer epochs compared to the other methods. This quick convergence suggests that the GM3D module helps the model focus on important features in the data more effectively, speeding up the learning process.

The Point-MAE model, while still effective, achieves a lower accuracy and takes more epochs to get there, indicating that it learns more slowly. On the other hand, the Point-MAE+GM3D* version (integrating the GM3D method with Point-MAE alongside $\mathcal{L}^{rec_p}$ and $\mathcal{L}^{GC}$) is also effective but doesn't converge as quickly as the Point-MAE+GM3D, showing the importance of knowledge distillation alongside GM3D module. These findings highlight the practical benefits of adding the GM3D module to the Point-MAE framework. By helping the model learn faster and more reliably, the GM3D module not only improves the model's overall performance but also reduces the time needed to achieve high accuracy.

**Fine-tuning Phase.** Fig. 6 displays the fine-tuning accuracy on the OBJBG dataset, providing clear evidence of the benefits brought by integrating our GM3D module with Point-MAE. The results reveal that the

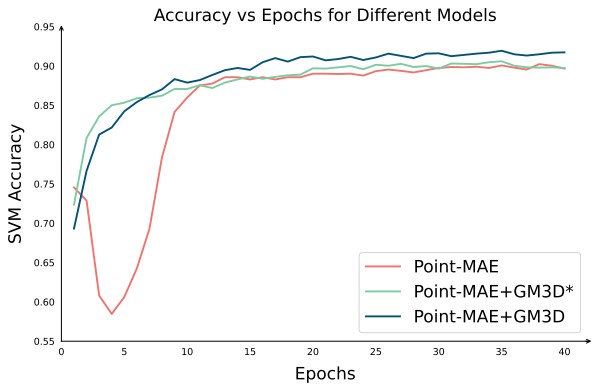
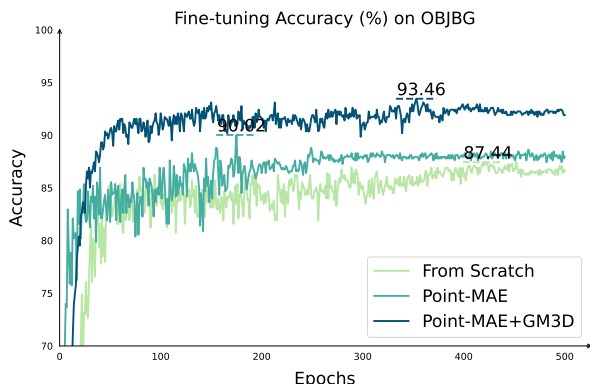

Figure 5: Comparison of convergence speed during the training phase (Point-MAE).

Figure 6: Fine-tuning vs. Training from Scratch on SacnObjectNN (Point-MAE).

Point-MAE+GM3D model not only achieves the highest accuracy but also maintains this improvement consistently over the course of 400 epochs. This consistent performance highlights the stability and effectiveness of the GM3D module in guiding the model to learn more relevant features from the data.

## 4.5 Ablation Study

Our ablation study focuses on the incremental improvements offered by our proposed method GM3D when integrated with the original Point-MAE framework. The original Point-MAE serves as our baseline, using the Chamfer loss for self-supervised learning and setting a performance benchmark on subsequent pretraining and fine-tuning tasks.

**GM3D.** Initially, we integrate the GM3D method with Point-MAE alongside $\mathcal{L}^{rec_p}$ and $\mathcal{L}^{GC}$. As reported in Table 7, this combination, termed Point-MAE+GM3D$^*$, shows a clear improvement over the baseline model by achieving higher pretraining SVM evaluation metrics on ModelNet40 and better fine-tuning results on ScanObjectNN (OBJ-ONLY). This supports the idea that a training focus on more geometrically complex patches contributes to improved model generalization.

Building on this structure, we enhance the performance by incorporating knowledge distillation alongside the GM3D module. The improved model, Point-MAE+GM3D, which employs three distinct loss functions, not only outperforms the baseline Point-MAE but also shows further improvement over the Point-MAE+GM3D$^*$ approach that utilizes only $\mathcal{L}^{rec_p}$ and $\mathcal{L}^{GC}$. This advancement validates the effectiveness of our knowledge distillation strategy, which focuses on accurate reconstruction while also capturing the complex geometric interrelations in the data. The various impacts of knowledge distillation are further explored in Table 8.

**Maximum Hard Patch Ratios.** The data presented in Table 2 offer insights into the ablation study focusing on different hardness ratios, denoted by $A$, within the context of point cloud modeling. It is noteworthy that the inclusion of GM3D enhances the performance across different $A$ settings when compared to the original model, with the highest performance observed at a 50% hardness ratio, where the OBJ-ONLY score reaches 90.36%.

Table 7: Comparison of Point-MAE, and Point-MAE+GM3D on Pretraining (SVM) and Fine-tuning (OBJ-ONLY) Tasks. '*' stands for our method without $\mathcal{L}^{rec_f}$.

| Model | Loss Function | SVM ModelNet40 | OBJ-ONLY |
|---|---|---|---|
| Point-MAE | $\mathcal{L}^{rec_p}$ | 91.05 | 88.29 |
| Point-MAE $+$ GM3D$^*$ | $\mathcal{L}^{rec_p} + \mathcal{L}^{GC}$ | **91.45** | **89.50** |
| Point-MAE $+$ GM3D | $\mathcal{L}^{rec_p} + \mathcal{L}^{rec_f} + \mathcal{L}^{GC}$ | **92.30** | **90.36** |

Table 8: Ablation study on different components of our method based on Point-MAE

| | Input $\to$ F | | $\mathcal{L}$ | | $\mathcal{L}^{rec} \to \mathcal{L}^{GC}$ | | $GM3D^t$ | | |
| | $X^v$ | $X$ | $\mathcal{L}^{rec_f}$ | $\mathcal{L}^{rec_p}$ | $\mathcal{L}^{rec_f}$ | $\mathcal{L}^{rec_p}$ | $\mu$ | $\mathcal{L}^{rec_f}$ | OBJ-ONLY |
|---|---|---|---|---|---|---|---|---|---|
| $a$ | ✓ | | ✓ | | ✓ | | ✓ | $\mathcal{E}^f \to \mathcal{D}^s$ | 89.32 |
| $b$ | ✓ | | ✓ | ✓ | ✓ | ✓ | ✓ | $\mathcal{E}^f \to \mathcal{D}^s$ | 90.18 |
| $c$ | | ✓ | ✓ | ✓ | | | ✓ | $\mathcal{E}^f \to \mathcal{D}^s$ | 89.67 |
| $d$ | | ✓ | | ✓ | | | ✓ | $\mathcal{E}^f \to \mathcal{D}^s$ | 89.50 |
| $e$ | | ✓ | ✓ | ✓ | ✓ | ✓ | | $\mathcal{E}^f \to \mathcal{D}^s$ | 89.33 |
| $f$ | | ✓ | ✓ | ✓ | ✓ | ✓ | ✓ | $\mathcal{E}^f \to \mathcal{E}^s$ | 89.15 |
| $g$ | | ✓ | ✓ | ✓ | ✓ | ✓ | ✓ | $\mathcal{E}^f \to \mathcal{D}^s$ | **90.36** |

**Additional Configurations.**  In Table 8, which details our ablation study, we investigate the various configurations of our proposed method. The 'Input' column pertains to the input utilized by the knowledge-teacher network; it specifies whether complete data $X$ is provided or only partial data $X^v$ are used. The second column, denoted by $\mathcal{L}$, encompasses both $\mathcal{L}^{rec_f}$ and $\mathcal{L}^{rec_p}$, representing the reconstruction loss functions in two spaces. In the third column, we analyze the impact of the chosen loss functions serving as the ground truth for $\mathcal{L}^{GC}$, which is our geometric complexity loss. As can be seen, the performance is enhanced by the geometric complexity guidance, which is informed by the feature-level knowledge distillation (rows c, and g). The subsequent column considers the influence of momentum, a parameter linked to the performance of GM3D$^t$. In the fifth column, we evaluate the impact of implementing $\mathcal{L}^{rec_f}$ on the interactions between various components of GM3D$^s$ and F. As evidenced by the results, each setting has been systematically varied to assess its effect on the final performance metric, OBJ-ONLY, demonstrating the significant contributions of each component to the model's learning efficacy.

## 5  Conclusion

We presented a geometrically-informed masked selection strategy for point cloud representation learning. Our GeoMask3D (GM3D) approach leverages a teacher-student model to find complex-to-reconstruct patches in the point cloud, which are more informative for learning robust representations. A knowledge distillation is further proposed to transfer rich geometric information from the teacher to the student, thereby improving the student's reconstruction of masked point clouds. Comprehensive experiments on several datasets and downstream tasks show our method's ability to boost the performance of point cloud learners.

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
