# GeoMask3D: Geometrically Informed Mask Selection for Self-Supervised Point Cloud Learning in 3D - Supplementary Material

Reviewed on OpenReview: `https://openreview.net/forum?id=Yk7GUlJwGa`

## 1 Implementation Pipeline

We used PyTorch to implement the core functionalities of our approach. The codebase is structured into two primary parts: *main-pretrain* and *main-finetune*.

**Pretraining Phase (*main-pretrain*).** In the *main-pretrain* section, we focus on the initial training phase of our models. In this phase, which is crucial to establish a robust foundation for our models, we first load the ShapeNet dataset and then apply our methods to train the models. algorithm 1 summarizes the training steps for GeoMask3D (GM3D) and includes the pseudocode for the reconstruction of point clouds, generation of geometric complexity, and the distillation of knowledge within the feature space.

**Finetuning Phase (*main-finetune*).** Once the pretraining is complete, we proceed to the *main-finetune* part. In this stage, only the encoder of student $\mathcal{E}^s$ from the pretrained models is carried forward. The output of this phase is directly responsible for the experimental results presented in our paper.

To ensure complete transparency and reproducibility of our results, we have made all relevant materials publicly available. This includes:

- The full source code for both *main-pretrain* and *main-finetune* phases.

- All log files containing the detailed results of our experiments

- Pretrained models for both pretraining and finetuning stages.

All these resources can be accessed through our GitHub repository. This repository includes everything needed to understand our code, covering all aspects of the implementations and the reproduction of the results. Moreover, the specifics of our network's hyperparameters for the pretraining and fine-tuning phases are comprehensively detailed in Table 1. Additionally, the pre-trained model for the Knowledge Teacher is selected based on the baseline methods, Point-MAE and Point-M2AE.

## 2 Additional Visualization

**Geometric Complexity.** Fig. 1 illustrates the Geometric Complexity (GC) analysis of randomly selected point clouds from the ShapeNet dataset. This illustration highlights the model's capability to assess GC at the patch level, where the red points denote areas of high GC, while the blue points indicate areas of low GC.

**Reconstructed Points.** To elucidate the capabilities of Masked Autoencoders (MAEs) in processing point cloud data, Fig. 2 provides a visual sequence involving the original input, the intermediate masking phase, and the reconstructed output. The first column, titled "Input Point Cloud", displays the entirety of the point cloud data, illustrating the initial condition before any processing. The subsequent column, "Masked Point Cloud", reveals only the points that remain visible after a portion of the data has been masked. The final

---

**Algorithm 1** Pseudo-Code of GM3D in a PyTorch-like Style

---

# teacher inference
$-, GC^t = GM3D^t(X)$
# curriculum patch selection
$M = \text{Mask-Generation}(GC^t, N^{sel})$
# student forward to compute objectives
$X^{rec}, f^{rec_s}, GC^s = GM3D^s(X^v)$
# knowledge teacher (frozen graph)
$f^{rec_f} = \mathcal{E}^f(X)$
# compute losses
# feature-space
$\mathcal{L}^{rec_f} = MSE(f^{rec_s}[M], f^{rec_f}[M])$
# point-space
$\mathcal{L}^{rec_p} = \text{Chamfer}(X^{rec}[M], X[M])$
# both spaces
$\mathcal{L}^{rec} = \mathcal{L}^{rec_p} + \mathcal{L}^{rec_f}$
# geometric complexity
$\mathcal{L}^{GC} = \text{DRC}(\mathcal{L}^{rec}, GC^s, M)$
# final loss
$\mathcal{L} = \mathcal{L}^{rec} + \mathcal{L}^{GC}$
return $\mathcal{L}$

---

column, "Reconstructed Point Cloud", demonstrates the model's ability to infer and restore the masked parts of the point cloud. The visual comparison in Fig. 2 distinctly highlights the high accuracy of the reconstructed points, underscoring the efficacy of our proposed method. It is noteworthy that these visual results were obtained using Point-MAE+GM3D. For a fair and consistent comparison, the mask ratio used here is like Point-MAE (60%).

## 3   Additional Analyses

**Pretraining Phase.** In Fig. 3, we illustrate the progression of SVM Accuracy throughout the pretraining phase on the ModelNet40 dataset. The red curve represents the Point-M2AE model, while the blue curve denotes the Point-M2AE enhanced with our GM3D method. It is evident from the graph that the incor-

Table 1: Hyperparameter configuration

| Config | Value |
|---|---|
| Optimizer | AdamW |
| Base learning rate | 1e-3 |
| Weight decay | 0.05 |
| Momentum | $\beta_1, \beta_2 = 0.9, 0.95$ |
| $\alpha$ | 1.0 |
| $\beta$ (After epoch 15) | 1000.0 |
| $\gamma$ (After epoch 15) | 10.0 |
| Batch size (Point-MAE+GM3D) | 256 |
| Batch size (Point-M2AE+GM3D) | 128 |
| Learning rate schedule | cosine decay |
| Pre-training epochs | 400 |
| Fine-tuning epochs | 500 |
| Augmentation (pretraining) | random scaling and translation |
| Augmentation (finetuning) | random rotation |

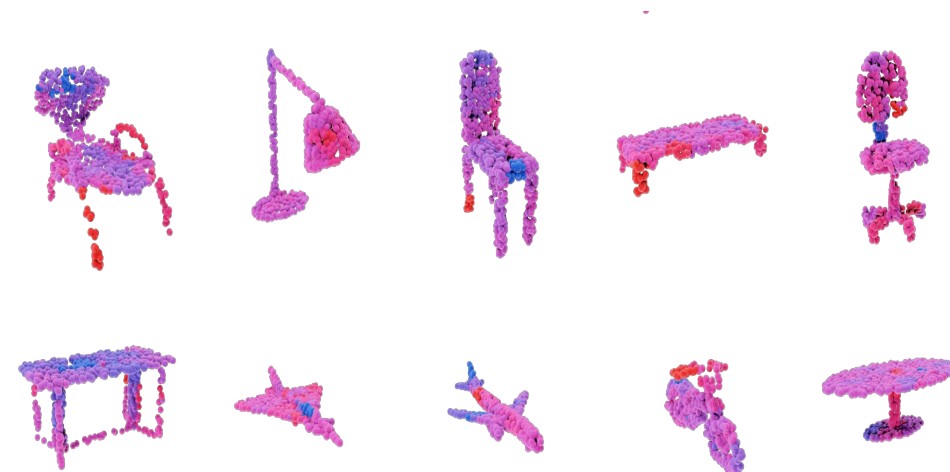

Figure 1: Visualization of GC values on diverse point clouds from the ShapeNet dataset.

poration of GM3D leads to a substantial improvement in SVM accuracy, reflecting the model's enhanced classification performance. A key observation is the accelerated convergence rate of the GM3D-augmented model; it achieves a rapid increase in accuracy within the initial epochs, demonstrating not only the efficacy of GM3D in facilitating faster learning but also indicating an enhanced ability to generalize from the training data.

**Fine-tuning Phase.** Fig. 4 displays fine-tuning accuracy on the OBONLY dataset, revealing that the integration of our GM3D module with Point-M2AE leads to the highest accuracy. Compared to the baseline Point-M2AE and the model trained from scratch, Point-M2AE+GM3D demonstrates a more significant improvement and exhibits less variability.

**Integration of GeoMask3D into Point-FEMAE.** In this section, we present an experimental evaluation of integrating our GeoMask3D approach into the Point-FEMAE **?** framework. Point-FEMAE employs two types of masking strategies: global masking and local masking. To enhance its capability, we modified the network by incorporating the geometric complexity decoder $\mathcal{D}_{GC}^s$ and replacing the original random global masking strategy with our geometrically guided masking technique.

While Point-FEMAE also utilizes local masking, where Euclidean distances guide the selective masking of tokens related to meaningful parts of the object, this strategy already contributes to capturing geometric structures effectively. However, this local masking approach may overlap with the objectives of our GeoMask3D, potentially reducing its distinct impact. Despite this, integrating these two methods provides valuable insights into the effectiveness of geometrically guided masking.

To assess the effectiveness of our approach, we pre-trained Point-FEMAE+GM3D on the ShapeNet dataset and evaluated it on the OBJ-ONLY dataset. The results, presented in Table 2, demonstrate that our modification improves performance. For comparison, we reproduced the baseline results of Point-FEMAE on this dataset using the original code and the pre-trained model available in the official repository.

Table 2: Point-FEMAE with and without GeoMask3D on the OBJ-ONLY dataset

| Method | OBJ-ONLY |
|---|---|
| ss Point-FEMAE | 92.08 |
| ss **Point-FEMAE + GM3D** | **92.77** |

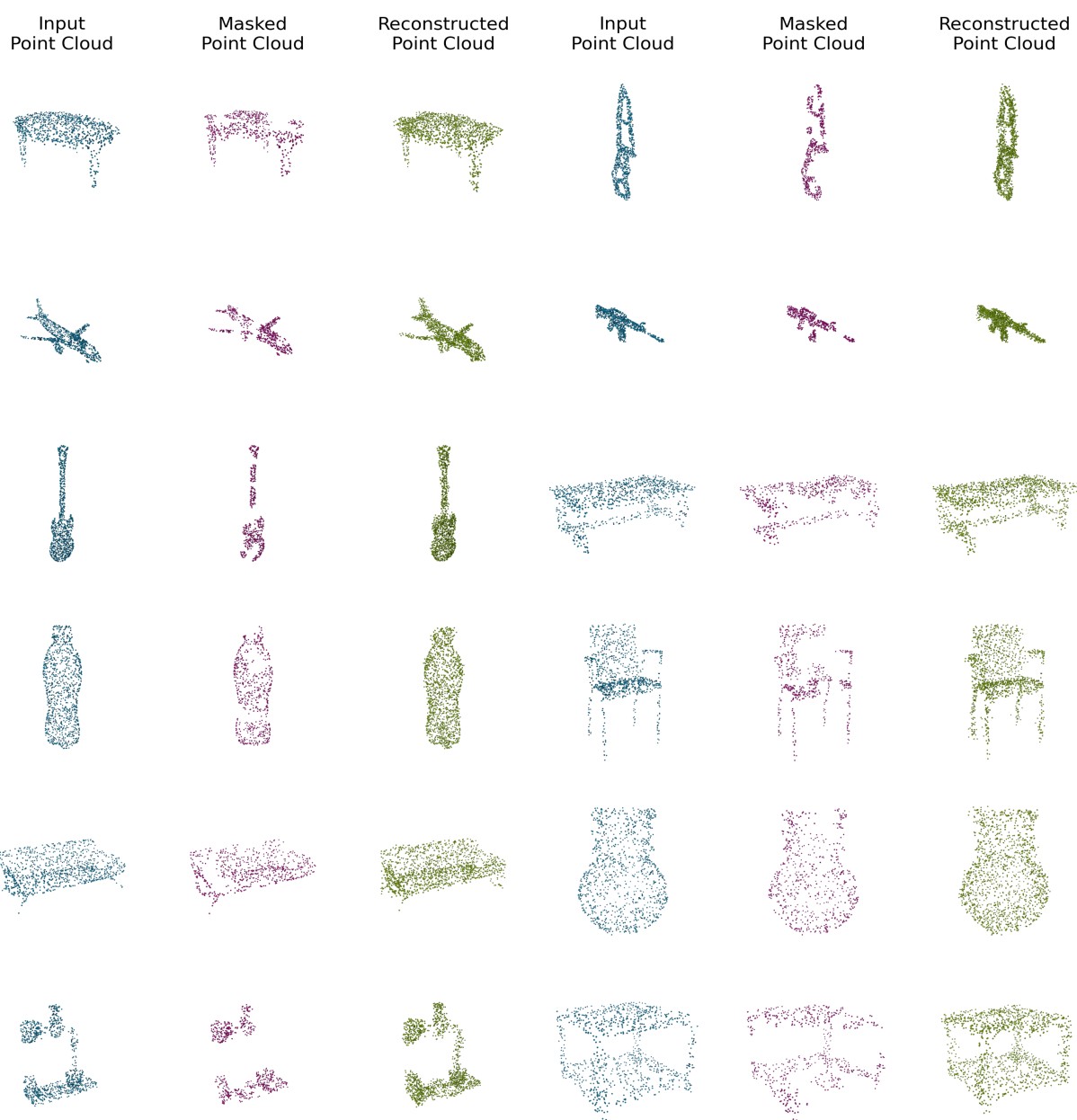

Figure 2: Reconstruction results on the ShapeNet dataset.

**Impact of Geometric-Guided Masking.** To further analyze the contribution of geometric-guided masking to the performance improvements observed in our method, we conducted an ablation study to isolate its effect from the GC prediction task.

To investigate the isolated impact of GC prediction, our method is pre-trained using the Point-MAE backbone with GC prediction but without geometric-guided masking. Instead of our masking strategy, we employed random masking. The objective of this experiment was to determine whether GC prediction alone contributes to the performance gains or if the synergy with geometric-guided masking is essential. We evaluated the pretrained model on the OBJ-ONLY dataset, as presented in Table 3.

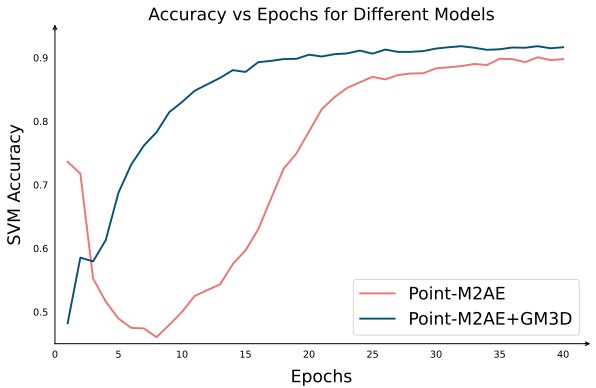

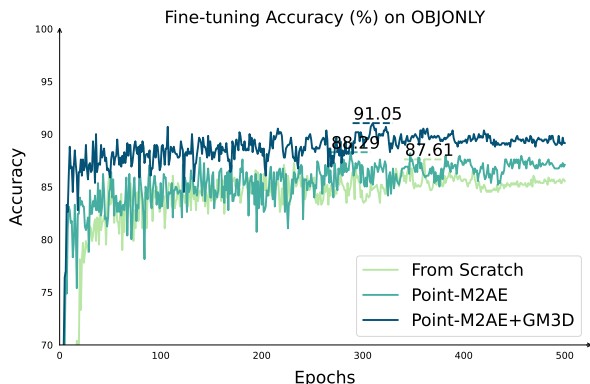

Figure 3: Comparison of convergence speed during the training phase (Point-M2AE).

Figure 4: Fine-tuning vs. Training from Scratch on SacnObjectNN (Point-M2AE).

As shown in the results, the model trained with both GC prediction and geometric-guided masking outperforms the one using GC prediction alone. This highlights the necessity of integrating geometric-guided masking with GC prediction to achieve optimal performance.

## 4 Time Analysis

In this section, we provide a detailed analysis of the computational resources required by our method compared to the baseline Point-MAE backbone. All experiments were conducted under identical hardware settings using an NVIDIA A6000 GPU with a batch size of 128.

### 4.1 Pre-Training

To evaluate the computational overhead introduced by our approach, we measured the pre-training time per epoch. The results indicate that:

- Point-MAE requires approximately 2.8 minutes per epoch.

- Our method requires 4.3 minutes per epoch.

The additional 1.5 minutes per epoch in our method is not solely due to the inclusion of the knowledge teacher model (which remains frozen during training), but also results from the computational procedures involved in predicting geometric complexity.

### 4.2 Downstream Tasks

As highlighted in the main paper, in downstream tasks, only the encoder of the student model is utilized. Since this encoder is identical to the one used in the baseline methods, our approach maintains computational efficiency during downstream inference. Thus, our method incurs no additional computational burden compared to baseline methods in downstream tasks.

Table 3: Ablation study on the impact of Geometric-Guided Masking

| Method | OBJ-ONLY |
|---|---|
| SS Point-MAE + GC Prediction + Random Masking | 89.67 |
| SS **Point-MAE + GC Prediction + Geometric-Guided Masking** | **90.36** |