# OpenReview forum: "GeoMask3D: Geometrically Informed Mask Selection for Self-Supervised Point Cloud Learning in 3D"
_TMLR — Accepted by TMLR_

### Review · Reviewer_JYxP · 2024-09-17

**Summary Of Contributions:**

This paper proposes a novel self-supervised method for point cloud processing. The main idea is to mask point patches in an easy-to-hard way and use a teacher-student framework to additionally "supervise" the student model. Extensive experiments are conducted to validate the proposed method.

**Audience:**

Yes

**Broader Impact Concerns:**

None.

**Claims And Evidence:**

Yes

**Requested Changes:**

Please see weaknesses 1,2,3.

In Figure 1, it is better to add scale/number indications.

**Strengths And Weaknesses:**

Strengths:
1. The motivation of this paper is clear. By utilizing 3D geometry information, the paper designs a novel masking strategy. From experiments, this masking strategy seems useful.
2. The proposed teacher-student framework also seems effective.
3. The paper is well-formatted and easy to follow.
4. Extensive experiments are conducted to show the effectiveness of the proposed method.

Weaknesses:
1. In the experiment part, the description of the teacher model is missing. Which model is used as a teacher?
2. It seems there are unfair comparisons between the proposed method and Point-MAE. The authors pre-train & fine-tune the model for 400&500 epochs respectively, while Point-MAE trains 300&300 epochs.
I understand that the proposed method uses an easy-to-hard masking strategy and might take a longer time to converge during pre-training, but I think it is better to finetune the same epochs while making comparisons.
3. I was curious about the computational resource comparison with baseline methods during pre-training as an additional teacher model is used.

---

> ### Author Response · Authors · 2024-09-19
>
> We would like to express our sincere gratitude for your thorough review of our paper. Your insightful comments have been invaluable in refining our work. We have addressed all your points in turn below.
>
> **1. In the experiment part, the description of the teacher model is missing. Which model is used as a teacher?**
>
> **Answer:** Regarding the Teacher model, it is selected based on the baseline methods we employed—specifically, Point-MAE and Point-M2AE. When using Point-MAE as the baseline, we utilize its encoder and decoder, along with an additional head consisting of an MLP for predicting geometric complexity, as the Teacher model. Similarly, when the baseline is Point-M2AE, we use the same structure with Point-M2AE as the Teacher model.
>
> As for the Knowledge Teacher, the model used is also selected based on the baseline methods—Point-MAE and Point-M2AE. In this case, only the encoder is used as the Knowledge Teacher.
>
> **2. It seems there are unfair comparisons between the proposed method and Point-MAE. The authors pre-train-fine-tune the model for 400-500 epochs respectively, while Point-MAE trains 300-300 epochs. I understand that the proposed method uses an easy-to-hard masking strategy and might take a longer time to converge during pre-training, but I think it is better to finetune the same epochs while making comparisons.**
>
> **Answer:**
> Thank you for bringing this important point to our attention. To ensure a fair comparison, as you suggested, we have run the experiments by fine-tuning our model with both backbones (Point-MAE and Point-M2AE) for 300 epochs on the ModelNet-40 and ScanObjectNN datasets. This is consistent with the fine-tuning approach we used for the segmentation and few-shot tasks, where our method is also fine-tuned for 300 epochs. For the ModelNet-40 dataset, we obtained the same accuracy of 94.20% with the Point-MAE backbone, which suggests that fine-tuning beyond 300 epochs does not improve performance on this dataset. For the ScanObjectNN dataset, we have updated the results in Table 1 of this rebuttal. As you can see, some results remain the same, indicating that the model does not require 500 epochs of fine-tuning to achieve optimal performance. The random seed used in our experiments also influences the consistency in results. The revised results demonstrate that our method still achieves superior performance compared to Point-MAE and Point-M2AE, even when fine-tuned for the same number of epochs. In the final version, all fine-tuning results are based on 300 epochs to align with the baseline methods' settings.
>
> | Method                           | OBJ-BG | OBJ-ONLY | PB-T50-RS |
> |-----------------------------------|--------|----------|-----------|
> | Point-MAE       | 90.02  | 88.29    | 85.18     |
> | **Point-MAE + GM3D**              | **93.11** | **90.36** | **88.30** |
> | Point-M2AE      | 91.22  | 88.81    | 86.43     |
> | **Point-M2AE + GM3D**             | **94.14** | **90.70** | **87.64** |
>
> **3. I was curious about the computational resource comparison with baseline methods during pre-training as an additional teacher model is used.**
>
> **Answer:**
> To address your question, we analyzed the computational resources required by our method compared to the baseline Point-MAE backbone under identical settings—a GPU A6000 and a batch size of 128. We measured the pre-training time per epoch and found that Point-MAE requires approximately **2.8** minutes per epoch, while our method takes about **4.3** minutes per epoch.
>
> The additional overhead of **1.5** minutes per epoch in our method is not solely due to the inclusion of the knowledge teacher model (which remains frozen during training), but also results from the procedures involved in predicting geometric complexity.
> We believe that this moderate increase in computational time is justified by the significant performance improvements our method achieves over the baseline. Additionally, as we mentioned in the paper, in downstream tasks, only the encoder of the student model—which is completely identical to the encoder of the baseline methods—is used. Therefore, in terms of computational resources during downstream tasks, our method is equivalent to the baseline methods.
>
> **4. In Figure 1, it is better to add scale/number indications.**
>
> **Answer:**
> We will revise Figure 1 accordingly in the final version of the paper.

---

> ### Comment · Reviewer_JYxP · 2024-09-23
> **More concerns**
>
> Thanks to the authors for solving my previous concerns.
>
> I admit the novelty and clear motivation of this framework. However, I have one more concern.
>
> In the proposed framework, it seems we need a pre-trained teacher model (like Point-MAE) to determine the geometry complexity and to "supervise" the student model. It sounds like the authors propose a self-supervised learning method that requires using another self-supervised learning method in advance. If we train from scratch, Point-MAE takes 2.8 mins/epoch while the proposed method takes 2.8 + 4.3 mins/epoch on average. In that way, the overall framework is a bit cumbersome.
>
> But overall, I still find this paper interesting and novel. The insight of the proposed method is inspiring.

---

> ### Author Response · Authors · 2024-09-23
>
> Thank you for your positive feedback. Regarding your concern, we use pretrained models when utilizing known backbones such as Point-MAE and Point-M2AE, as their pretrained models are readily available. However, as you correctly pointed out, when introducing a new backbone, additional pretraining time is required. We acknowledge this overhead and will mention it in the limitations section of the paper.

---

### Review · Reviewer_o5sU · 2024-11-11

**Summary Of Contributions:**

**Summary**:
This paper proposes the geometrically informed mask selection strategy (GeoMask3D) to improve the performance of MAE for point clouds. Specifically, GeoMask3D adopts a teacher-student model to focus on the regions with higher geometric complexity, leading to a more robust feature representation. Finally, GeoMask3D achieves marked improvements in classification, segmentation, and few-shot tasks.

**Audience:**

Yes

**Broader Impact Concerns:**

No any concerns on the ethical implications of the work.

**Claims And Evidence:**

Yes

**Requested Changes:**

1. Provide more detailed ablation studies about Geometric-Guided Masking to prove that the performance gain does not brougth by GC prediction task but by Geometric-Guided Masking.
2. Compare more improved MIM methods (e.g., Point-FEMAE).
3. More discussion and experiments are needed to compare MIM of MVP.

**Strengths And Weaknesses:**

**Strengths**:
1. GeoMask3D foucs on the regions with higher geometric complexity, which makes sense.
2. GeoMask3D achives superior performance compared with SOTA baseline.

**Weaknees**:
1. The definition of GT and GC GT is not clear, which makes it difficult to understand.
2. I suggest that authors provide more detailed ablation studies about Geometric-Guided Masking to prove that the performance gain does not brougth by GC prediction task but by Geometric-Guided Masking.
3. I think it is not sufficient to compare only PointMAE and PointM2AE. There are many improved MIM methods (e.g., Point-FEMAE [1]).
4. The feature-level knowledge distillation is a common technique (e.g., MVP [2]). More discussion and experiments are needed to compare MIM of MVP.

[1] Towards Compact 3D Representations via Point Feature Enhancement Masked Autoencoders

[2] MVP: Multimodality-guided Visual Pre-training

---

> ### Author Response · Authors · 2024-11-25
> **1. Provide more detailed ablation studies about Geometric-Guided Masking to prove that the performance gain does not brougth by GC prediction task but by Geometric-Guided Masking.**
>
> Thank you for your insightful feedback and suggestions. Your input has significantly contributed to improving the clarity and comprehensiveness of our work. We sincerely appreciate your time and effort in reviewing our paper.
>
> In our approach, Geometric Complexity (GC) is first predicted by the student network, and this information is progressively transferred to the teacher network using an Exponential Moving Average (EMA) mechanism. The teacher network then predicts the GC of the original point cloud, identifying tokens or patches that are inherently challenging to reconstruct based on a specific threshold. These challenging tokens are subsequently selected for masking in the student network during the next iteration.
>
> If we were to exclude the GC prediction loss, we could rely solely on the reconstruction loss (in both point-space and feature-space) to identify tokens that are hard to reconstruct. This could theoretically allow us to mask such tokens in the next iteration and observe the isolated effect of Geometric-Guided Masking. However, this approach introduces a critical limitation:
>
> In all previous MAE frameworks for point clouds and images, such as Point-MAE, Point-M2AE, I2P-MAE, IAE, and MAE, the reconstruction loss is calculated only on the masked tokens. Consequently, without the GC prediction loss, we can only evaluate which of the initially and randomly masked tokens are easy or hard to reconstruct. This creates a static selection of masked tokens, which does not allow for dynamically identifying and targeting genuinely hard patches. This results in a static selection of masked tokens (fixed tokens), preventing the dynamic identification and targeting of truly challenging patches. Consequently, implementing our method under such conditions is not feasible.
>
> For this reason, GC prediction and Geometric-Guided Masking are fundamentally interdependent, and using one without the other would not yield meaningful results. GC prediction provides the foundation for dynamically identifying challenging tokens, while Geometric-Guided Masking ensures the model focuses on these tokens to enhance learning. Together, they form a cohesive framework that drives the observed performance improvements.
>
> To address and respond to your concern, we pretrained our method using the backbone of Point-MAE with the $L^{GC}$ loss. However, in this evaluation, we did not use geometrically guided masking for token masking; instead, we employed random masking. The objective of this evaluation was to isolate the effect of $L^{GC}$ without the influence of geometrically guided masking.
> We then evaluated the pretrained model on the downstream task using the OBJ-ONLY dataset. The model achieved an accuracy of **89.67%**, whereas using both $L^{GC}$ and geometrically guided masking resulted in an accuracy of **90.36%**. As shown in the results, the performance is lower when only $L^{GC}$ is used compared to the case where both $L^{GC}$ and geometrically guided masking are utilized. This highlights the importance of combining $L^{GC}$ with geometrically guided masking to achieve optimal performance.
>
> Thank you for your suggestion. It is a valuable ablation, and we will include it in the results section of the final version of our paper.

---

> ### Author Response · Authors · 2024-11-25
> **2. Compare more improved MIM methods (e.g., Point-FEMAE).**
>
> We reviewed the Point-FEMAE paper and understood that this method employs two types of masking strategies: global masking and local masking. To incorporate our approach (GeoMask3D), we modified Point-FEMAE by adding the geometric complexity decoder $D^{s}_{GC}$ to the network and replacing the global masking strategy, which is originally random, with our geometrically guided masking strategy.
>
> However, it is important to note that this method uses local masking alongside global masking. In local masking, using Euclidean distances, tokens related to meaningful parts of the object are selectively masked. This strategy helps the model capture the geometric information of the object more effectively. For instance, as shown in the Fig 2 in the paper, masking meaningful local regions of a plane (e.g., the wings and fuselage) ensures the model learns critical geometric structures.
>
> While this local masking strategy enhances geometric learning, it may reduce the impact of our GeoMask3D approach by overlapping with its objectives. Nonetheless, the combination of these methods provides valuable insights into the effectiveness of geometrically guided masking.
>
> As for your suggestion, we pre-trained Point-FEMAE+GeoMask3D on the ShapeNet dataset and evaluated it on the OBJ-ONLY dataset. The results in Table 2 show that our approach successfully improves the performance. Regarding the baseline results of Point-FEMAE, we reproduced them on this dataset using the original code and the pre-trained model available on their official GitHub repository.
>
> | Method                  | OBJ-ONLY |
> |-------------------------|:--------:|
> | Point-FEMAE            |   92.08  |
> | **Point-FEMAE + GM3D** | **92.77** |

---

> ### Author Response · Authors · 2024-11-25
> **3. More discussion and experiments are needed to compare MIM of MVP.**
>
> We reviewed the MVP paper thoroughly and gained valuable insights into its approach and contributions. As highlighted in MVP, CLIP is indeed a powerful vision-language pre-trained model that effectively combines visual and semantic representations. However, as you may know, there is currently no CLIP pre-trained model specifically for 3D point clouds. All existing work in this domain employs projections of point clouds into 2D images, which are then fed into the visual backbone of CLIP, as demonstrated in works like Point-CLIP [1] and Point-CLIP-V2 [2].
>
> These papers show that converting 3D point clouds into 2D images reduces the quality of spatial structure, which lowers accuracy. The loss of fine-grained 3D geometry during the projection process makes it challenging to achieve optimal performance for downstream 3D tasks.
>
> Given these observations and the results of our method, we believe that directly implementing the MVP approach, which relies on a 2D pre-trained backbone and multimodal alignment, would not be beneficial for our 3D-focused framework. Instead, our approach is designed to leverage the distinct characteristics of 3D point clouds in their original form, bypassing the limitations of projection-based methods.
>
> We appreciate your suggestion and agree that integrating multimodal guidance into 3D MIM approaches holds promise. However, we believe it requires either a pre-trained multimodal model specifically designed for 3D data or a novel adaptation that better preserves 3D geometric fidelity. We will explore this direction further in future work.
>
> As shown in Table, our method, which is pre-trained directly on 3D point clouds, demonstrates superior performance across all benchmarks compared to CLIP-based methods (Point-CLIP and Point-CLIP-V2). The CLIP models, trained on 400M images and applied to point clouds via projection techniques, struggle to maintain the spatial fidelity and fine-grained geometry inherent to 3D data, resulting in significantly lower accuracy in downstream tasks.
>
> | Method                  | ModelNet40 | OBJ-BG  | OBJ-ONLY | PB T50 RS |
> |-------------------------|:----------:|:-------:|:--------:|:---------:|
> | Point-MAE              |   93.80    |  90.02  |   88.29  |   85.18   |
> | **Point-MAE + GM3D**   | **94.20**  | **93.45** | **90.36** | **88.30** |
> | Point-Clip             |   23.78    |  19.28  |   21.34  |   15.38   |
> | Point-Clip-V2          |   64.22    |  41.22  |   50.09  |   35.36   |
>
>
> [1] Zhang, R., Guo, Z., Zhang, W., Li, K., Miao, X., Cui, B., ... & Li, H. (2022). Pointclip: Point cloud understanding by clip. In Proceedings of the IEEE/CVF conference on computer vision and pattern recognition (pp. 8552-8562).
>
> [2] Zhu, X., Zhang, R., He, B., Guo, Z., Zeng, Z., Qin, Z., ... & Gao, P. (2023). Pointclip v2: Prompting clip and gpt for powerful 3d open-world learning. In Proceedings of the IEEE/CVF International Conference on Computer Vision (pp. 2639-2650).

---

> ### Comment · Reviewer_o5sU · 2024-12-06
> **I recommend accepting this paper.**
>
> Thank you for the authors' response.
>
> I recognize the contribution and innovation of this paper. Besides, I encourage authors to include these additional experiments in the final version of this paper.
>
> Overall, I recommend accepting this paper.

---

> > ### Author Response · Authors · 2024-12-07
> >
> > We sincerely appreciate your positive feedback. We will ensure that all the new results are incorporated into the final version of the paper.

---

### Review · Reviewer_LqZs · 2024-11-11

**Summary Of Contributions:**

This paper aims to boost the MAE learning of point clouds by the proposed geometrically informed mask selection strategy. Specifically, GeoMask3D first focus on the regions with higher geometric complexity and adopts teacher-student framework to achieve more robust feature representation. GeoMask3D improves the performance of MAE baseline on classification, segmentation, and few-shot tasks

**Audience:**

Yes

**Claims And Evidence:**

Yes

**Requested Changes:**

1. The written should be improved to present the proposed geometrically informed mask selection strategy and does Geometric Complexity.
2.  Authors should provide more experiments on more datasets (e.g., S3DIS, ScanNet, KITTI, Waymo) to prove the effectiveness of the proposed method.
3. See the part of weaknesses

**Strengths And Weaknesses:**

Strengths:
1. GeoMask3D proposes the geometrically informed mask selection strategy to achieve better performance.
2. GeoMask3D achieves better performance compared with baseline.

Weaknees:
1. The written about proposed geometrically informed mask selection strategy should be more detailed, which is the core of this paper.
2. What does Geometric Complexity represent and how is its GT generated?
3. The proposed feature-level knowledge distillation is a feature distillation technique, which is widely used in many works. I don't think there are any innovation.
4. GeoMask3D only conduct all experiments on only simple classification datasets (ModelNet40, and ScanObjectNN) I think more datasets (e.g., S3DIS, ScanNet, KITTI, Waymo) are needed to prove the effectiveness of the proposed method.

---

> ### Author Response · Authors · 2024-11-25
> **1. The written about proposed geometrically informed mask selection strategy should be more detailed, which is the core of this paper. 2. What does Geometric Complexity represent and how is its GT generated?**
>
> Thank you for your insightful feedback and suggestions. Your input has significantly contributed to improving the clarity and comprehensiveness of our work. We sincerely appreciate your time and effort in reviewing our paper.
>
> We have revised the writing to improve the presentation of the proposed geometrically informed mask selection strategy and its relationship to Geometric Complexity. These changes have been incorporated into Sections 3.2, and 3.2.1 of the main paper. Please refer to these sections for the updated explanation and details.

---

> ### Author Response · Authors · 2024-11-25
> **3. The proposed feature-level knowledge distillation is a feature distillation technique, which is widely used in many works. I don't think there are any innovation.**
>
> In conventional feature knowledge distillation, your observation is correct—knowledge is typically transferred from a teacher network to a student network, with both networks often processing the same input data. However, our method introduces a significant difference from conventional approaches. In our feature-level knowledge distillation, the complete point cloud is fed into the teacher network, while only the partial (masked) point cloud is fed into the student network. This creates a unique learning dynamic where the student learns not only from the observed patches but also indirectly from the unobserved patches via the latent features distilled from the teacher.
>
> This unique setup enables the student network to benefit from the global geometric context provided by the teacher network, which is constructed from the complete point cloud. This interplay between the full and partial point clouds is not found in conventional feature distillation techniques, making our approach innovative and specifically designed for point cloud representation learning. An ablation study for this component is presented in Table 8 of the paper, specifically comparing rows (b) and (g).
>
> We have revised Section 3.3 of the main paper to address this point. Please refer to this section for the updated content.

---

> ### Author Response · Authors · 2024-11-25
> **4. GeoMask3D only conduct all experiments on only simple classification datasets (ModelNet40, and ScanObjectNN) I think more datasets (e.g., S3DIS, ScanNet, KITTI, Waymo) are needed to prove the effectiveness of the proposed method.**
>
> Following our baselines—Point-MAE, Point-M2AE, I2P-MAE, IAE, Point-GPT-S, ACT, and Point-BERT—our method is specifically designed for object-level point clouds and we have not claimed generalizability to scene-level or outdoor point clouds.
>
> We have evaluated our method on all standard benchmarks commonly used for object-level point cloud processing (similar to our baselines). It is important to note that the nature of object-level point clouds significantly differs from scene-level and outdoor point clouds; None of these methods, including ours, are typically evaluated on scene-level tasks.
>
> For example for the S3DIS dataset, If we fine-tune our pretrained model, which was pretrained on the ShapeNet dataset (object-level), we cannot utilize the RGB features of the dataset. This problem arises because the pre-trained network only supports three input channels, restricting us to using the coordinate features of each point (x, y, z). Given the complexity of these scene-level datasets, the accuracy drops significantly compared to state-of-the-art methods. To validate this, we conducted a preliminary experiment by fine-tuning Point-MAE and Point-MAE+GM3D, both pretrained on the ShapeNet dataset (object-level), on the S3DIS dataset. The results (Point-MAE: **12.8%**, Point-MAE+GM3D: **12.8%**) confirm our observations regarding this limitation and the resulting drop in performance.

---

### Decision · Action_Editor_Vcm2 · 2025-01-22

**Recommendation:** Accept with minor revision

**Comment:**

This paper focuses on improving the application of the MAE method in 3D perception tasks. The core motivation is that existing random masking does not effectively utilize data information. The proposed GeoMask3D learns better features and enhances the performance of downstream tasks. The clear motivation behind leveraging 3D geometry information is well-executed, and the extensive experiments conducted validate the effectiveness of the proposed method. The teacher-student framework is also a strong addition to the approach. The paper does have some weaknesses, including insufficient detail on the core contributions, such as the geometrically informed mask selection strategy and the definition of key terms like Geometric Complexity and its ground truth. Despite these shortcomings, the paper's overall presentation is clear, and its proposed method shows promise in improving 3D perception tasks. The authors should carefully revise their paper with respect to the above issue in the minor revision so that the paper can reach the standard of publishment.

**Audience:**

Researchers who work on 3D perception and self-supervised learning may benefit from the findings of this paper. The proposed method may help downstream 3D perception tasks. The adaption on the masking strategy can offer new insights for future MAE works.

**Claims And Evidence:**

This paper claimed that their proposed masking technique, GeoMask3D, can outperform the commonly-used random masking strategy, and thus boost the effectiveness of self-supervised MAE paradigm. The claim is majorly verified by empirical results on 3D detection tasks and segmentation tasks. Consistent improvements are observed across various benchmarks. Additional visualization results also support the above claim.